# Cross-World Covert Channel on ARM Trustzone through PMU

**DOI:** 10.3390/s22197354

**Published:** 2022-09-28

**Authors:** Xinyao Li, Akhilesh Tyagi

**Affiliations:** Department of Electrical and Computer Engineering, Iowa State University, Ames, IA 50010, USA

**Keywords:** TrustZone, PMU, covert channel attack, IoT, OP-TEE

## Abstract

The TrustZone technology is incorporated in a majority of recent ARM Cortex A and Cortex M processors widely deployed in the IoT world. Security critical code execution inside a so-called secure world is isolated from the rest of the application execution within a normal world. It provides hardware-isolated area called a trusted execution environment (TEE) in the processor for sensitive data and code. This paper demonstrates a vulnerability in the secure world in the form of a cross-world, secure world to normal world, covert channel. Performance counters or Performance Monitoring Unit (PMU) events are used to convey the information from the secure world to the normal world. An encoding program generates appropriate PMU event footprint given a secret *S*. A corresponding decoding program reads the PMU footprint and infers *S* using machine learning (ML). The machine learning model can be trained entirely from the data collected from the PMU in user space. Lack of synchronization between PMU start and PMU read adds noise to the encoding/decoding ML models. In order to account for this noise, this study proposes three different synchronization capabilities between the client and trusted applications in the covert channel. These are synchronous, semi-synchronous, and asynchronous. Previously proposed PMU based covert channels deploy L1 and LLC cache PMU events. The latency of these events tends to be 100–1000 cycles limiting the bandwidth of these covert channels. We propose to use microarchitecture level events with latency of 10–100 cycles captured through PMU for covert channel encoding leading to a potential 100× higher bandwidth. This study conducts a series of experiments to evaluate the proposed covert channels under various synchronization models on a TrustZone supported Cortex-A processor using OP-TEE framework. As stated earlier, switch from signaling based on PMU cache events to PMU microarchitectural events leads to approximately 15× higher covert channel bandwidth. This proposed finer-grained microarchitecture event encoding covert channel can achieve throughput of the order of 11 Kbits/s as opposed to previous work’s throughput of the order of 760 bits/s.

## 1. Introduction

Processors and operating systems (OS) are designed to isolate two user processes or a kernel process and a user process. In a properly executed security architecture, information cannot flow between a privileged OS process and a user process, unless explicitly initiated by an OS process. Similarly, information from a user process *A* must not flow to another user process *B* unless the process *A* explicitly initiates it. However, despite the best design intentions, unintended operating system flaws do permit such information channels. These are called covert channels. Previous research has shown that covert channels can be constructed with a variety of existing system resources [1,2,3,4,5].

Most modern processors also include hardware supported trusted execution environments (TEE) such as Intel TXT, AMD SVM, ARM TrustZone, and Intel SGX. These secure subsystems ensure a more strict isolation between two computations of variable granularities; however, these TEEs have also been found to support covert channels significantly weakening their protection [6,7,8,9,10,11,12].

The best known TEEs are Intel’s Software Guard Extensions (SGX) and ARM’s TrustZone. Two recent trends make ensuring the integrity of TrustZone critical, (1) the internet usage on mobile devices exceeds the usage on PC since 2014 [13], (2) IoT population is exploding where ARM processors dominate. ARM TrustZone is a security extension with hardware-enforced isolation integrated into the main processor. ARM Cortex A and Cotex M cores are commonly deployed in IOT systems. The processor cores are divided into two sets of virtual cores, which are called normal world and secure world. In the normal world, client applications (CAs) run on the Rich Execution Environment (REE), whereas, in the secure world, only trusted applications (TAs) run on the Trusted Execution Environment (TEE). The protection of trusted applications from untrusted applications is provided throughout the entire architecture including hardware, software, and cryptographic isolation. This allows a programmer to entrust privileged information to the secure world. TrustZone is mostly used in Arm A-profile and Arm M-profile architecture with TEE implementations such as OP-TEE, Trusty, QSEE, and Kinibi.

While TrustZone is effective in providing isolation between a secure world and a normal world, a line of recent studies has shown that it is possible to build a covert side-channel. Previous covert channels are built based on cache side channel attacks by exploiting the variable access times to infer whether specific data have been accessed. In particular, Refs. [14,15,16,17,18,19] proposed cache side-channel attacks including Prime+Probe, Flush+Reload, Evict+Reload, and Flush+Flush on ARM TrustZone. Ref. [20] has shown the cache attacks on AES T-table implementations. The proposed attack modes in this paper are also instances of a family of covert channel attacks. While the previously proposed attacks use cache side-channel, this study uses the computation footprint readings from PMU counters to encode the covert channel bound information. We are not the first ones to use PMU to build a covert channel. The most relevant publication to our covert channel, proposed by [21] is called prime + count technique. The authors utilize the ARM Performance Monitoring Unit (PMU) feature named “L1/L2 cache refill events” to determine how many cache sets or lines have been occupied. However, unlike our attack, this prime+count attack is coarse-grained, which leads to low channel bandwidth of the order of 760 bits/s. Forcing the L1/L2 cache refill events to create a targeted PMU footprint takes significant amount of time due to inherent L1/L2 cache miss latencies. This study is able to achieve covert channel bandwidths of the order of 11 KBits/s due to the finer granularity of encoding.

As for cache side channel attack defenses, Refs. [22,23] suggests introducing “random noise” to the attackers’ cache observation, which reduces time difference between cache hit and cache miss with cache wait function designed by [24]. As a result, the attacks that are based on the cache co-occupancy techniques do not work anymore. The same performance can be obtained by another defense strategy proposed by [25], that is, by adding stub code before and after each protected function. Build on top of the Flush+Reload and Flush+Flush techniques, [26] also proposed a better performed cache covert channel uses less cache flush; however, all these defenses do not affect our covert channel, since this study only monitors encoding computation microarchitectural footprint through microarchitectural PMU event counters instead of cache accesses.

Overall, this paper makes the following contributions:This is the first covert channel built and trained through PMU readings from the user space. The PMU-based exfiltration encoding programs can model PMU footprint created through secure world execution in the normal world execution.This study presents three attack models with different levels of synchronization. The covert channel can achieve as high as 99.29% accuracy in inferring exfiltrated tokens from the secure world to the normal world.This study evaluates the covert channel bandwidth of the previous PMU cache events-based inference versus our covert channel-based on finer-grained microarchitecture level PMU events. Without the most time-consuming cache encoding and decoding process, our covert channel achieves 100 times higher throughput. Previous research explicitly generates L1/L2 cache level events to encode the leaked information into PMU event counters. The cache events takes 100 s of clock cycles. We instead use microarchitecture pipeline level events to encode the leaked information. These events take 10 s of clock cycles leading to a significant advantage in the covert channel bandwidth.This study shows that the existing covert channel mitigation techniques focused on cache side-channel attack do not mitigate our PMU-based covert channel. We do not need cache level event capture for our covert channel.

### 1.1. Background

ARM TrustZone technology or TrustZone for short is a system-on-chip approach used on ARMv6 and later architectures. The heart of TrustZone is the concept of separate computing resources for secure and non-secure worlds. Since ARM is widely deployed on the majority of mobile and microcontroller devices, the goal of TrustZone is to provide security for those platforms. The security is enforced through hardware, software, and cryptographic operation isolation mechanisms.

#### 1.1.1. TrustZone Hardware Architecture: Processors

The current ARM processors that support the ARM TrustZone technology are mainly in the Cortex-A and ARM-M family, consisting of ARM1176JZ(F)-S, Cortex-A8, Cortex-A9, Cortex-A9 MPCore, Cortex-M23, and Cortex-M33 [27]. Each of the physical processor cores in these designs provides two virtual cores, one is considered as non-secure or normal and the other is secure. There exists a mechanism to robustly context switch between them, known as monitor mode. The secure world is where everything runs when the processor state is secure, and the normal world is where everything runs when the processor is in the non-secure state. The context switch mechanism from normal to secure world is considered as a hardware barrier established to prevent normal world components from accessing secure world resources, while the secure world is not restricted by the barriers. In this project, we build and test our attack model on Raspberry Pi 3 Model B with quad-core Arm Cortex-A53 CPU, which provides ARM TrustZone exception states.

#### 1.1.2. TrustZone Software Architecture: Secure Subsystem

TrustZone software extensions, coupled with TrustZone hardware implementations, create secure and normal world isolation. Having a dedicated OS just for the secure world is the main focus of TrustZone software implementation. When it comes to implementing concurrent secure and non-secure/normal operating systems, a general operating system, such as Linux, Android, etc., would run in the normal world, and a security subsystem of the general OS, such as OP-TEE [28], would run in the secure world only. Besides OP-TEE, there are other popular software implementations of secure operating systems, including Google’s Trusty, Qualcomm’s QSEE, Trustonic’s Kinibi, etc. This study evaluates the proposed technique on the open source OP-TEE platform.

#### 1.1.3. TrustZone Cryptographic Operations

TrustZone-based TEE solutions can provide the functionality to support cryptographic operations. The cryptographic operations include secure key generation, key derivation, symmetric, and asymmetric cryptographic operations [29]. This study first implements cryptographic operations such as RSA or AES in a normal world application. Then, we train our machine learning prediction model based on the data collected from PMU readings of the normal world application. The data is collected from the normal world because it is not always feasible to collect the data in the secure world. This offers a more general paradigm where the training model is based on the mirrored state in the normal world, but works fairly well to infer the secure world states. This subverts many of the secure world defense mechanisms. The machine learning model trained on normal world state achieves high accuracy in secure world state predictions. The next step is to apply the trained model to detect secure key bits that are generated in the secure world.

#### 1.1.4. The Vulnerability of TrustZone

Previous studies have shown that the legitimate channels used to communicate between a normal world component and a secure world component are vulnerable [30]. The attacks proposed previously in Trustzone are mainly cache side channel attacks, while our attack utilizes the ARM Performance Monitor Unit to infer the secret contents in the secure world. To protect the legitimate channels’ communication in TrustZone, a number of strong monitors have been proposed. For example, SeCReT is proposed in [30] as a framework to protect the session key by restricting access to the communication channels on an access control list. This type of defense mechanism [30,31] is effective on previous cache side channels by monitoring the malicious activity in cache and communication channels. However, this study experimentally shows that such strong monitors including SeCReT cannot detect and disable the proposed covert channel.

### 1.2. Related Work

Cho et al. in [21] has proposed a Prime+Count cache side-channel attack by using a PMU feature named ”L1/L2 cache refill events”. It encodes a value *x* to be leaked from the secure world through cache line occupancy. The normal world part of the application first fills each cache line, priming, from its own address space. The encoding program in the trusted component of the application replaces some *x* of these lines from the trusted application’s address space. The normal world then reads these lines again to count the number of misses that should approximate *x*. The main drawback of any cache side channel-based covert channel, however, is overhead of encoding and decoding. The ARM Cortex A53 has 32KB L1 and 1MB L2 cache. For L1-based same core covert channel, of the order of 512 cache lines have to be primed. With an L1 miss penalty of order of 50 clock cycles, this comes to the order of 25,000 clock cycles. The encoding and decoding add roughly similar overhead leading to approximately 100,000 clock cycles overhead for each value communicated to transmit 2–7 bits. This leads to a theoretical covert channel throughput of the order of 10,000 bits/sond. Effective delivered throughput for this covert channel is of the order of 100 bits/sond. Additional losses occur in over encoding to protect against noise. A cross-core covert channel established through L2 co-occupancy would have an even higher overhead resulting in lower throughput.

Our work uses a computational module for encoding with an overhead below a 1000 clock cycles for encoding 4 bits at a time. This theoretically gives us a much higher throughput covert channel of the order of one million bits/s. We achieve of the order of 11 Kbits/s throughput.

Cho et al. [21] also did not demonstrate the results of the proposed attack against stronger monitors. To the best of our knowledge, the proposed covert channel attack in this project is the first attempt to build a fine-grained PMU based cross-world covert channel that cannot only run against strong monitors but is also able to extract the key bits. This study conducts a series of experiments to demonstrate the performance of the proposed techniques.

Most of the results in [21] appear to be based on a synchronous model, whereas this paper also considers more challenging semi-synchronous and asynchronous models.

## 2. Materials and Methods

### 2.1. Adversary Model

Our adversary model is an enhancement of the model used in [21]. The adversary wishes to create a covert channel from the secure world to the normal world to exfiltrate the secret data or key accessible only in the secure world. The obvious channels between the secure and normal worlds are monitored by SeCReT-like layer. The mere act of creating a covert channel needs an unmonitorable encoding method for bit groups. A symmetric decoding method in the normal world is also needed.

The encoding method in this work is a computational procedure where the PMU measured computation metrics between different encoded values are easily differentiable. Matrix multiplication serves as such an encoding method.

The encoding procedure for the covert channel merely pumps information into the covert channel. How the secret data/key is made available to the encoding method is an orthogonal issue, which is not addressed in this paper, just as it was not a focus in preceding covert channel works such as [21]. This study assumes that the client application (CA) invokes a service in the secure world trusted application (TA), such as encryption, that has access to a secret (such as RSA key). The service could either be provided by the adversary developed application, in which case, it is straightforward to infect it with the encoding function before returning to the CA. Or the encryption service could be built into the secure world. In this scenario, this study assumes that some other method has been used to infect the encryption library with the encoding function.

Note that the assumption that the secret keys available to the TA are only available during the cryptographic computation is stronger than the current capabilities of TrustZone. A root of trust for key management could generate the key from SRAM or some other kind of physical unclonable functions (PUFs) only when needed. Our covert channel could still be effective against such an adversary.

Our approach works both for the same core CA and TA as well as cross-core CA and TA.

### 2.2. Experimental Setup

In this section, we introduce three different types of attacks: synchronous access attack, semi-asynchronous access attack, and asynchronous access attack. The essential goal of these attack models is to steal sensitive information in the secure world, which is a secret key and is accessible to the TA running in the secure world.

For each attack model, we need to write two programs, one is called CA which runs in the normal world, another is called TA running in the secure world. When experiments start, CA is called in the normal world which further calls the TA to perform some calculations in the secure world by using the secret key as the input. The secret key bit values are encoded by executing an encoding program whose computational work is proportional to the encoded value. The computational work is measured through the PMU. This necessitates precise synchronization to start the PMU event counting in the normal world right before the encoding computation starts in the secure world. Similarly, PMU counter read in the normal world needs to be synchronized with the end of the encoding computation in the secure world. Different synchronization levels for the PMU initiation and read lead to different attack models illustrated in Figure 1. When both the PMU start and read are perfectly synchronized with the secure world encoding computation, we obtain a synchronous model, which is likely to be most precise. When only one of PMU start or read is synchronized, we obtain a semi-synchronous model. When both PMU start and read are asynchronous, we obtain the noisiest asynchronous model.

We begin with hardware and software choices for the experiment, and use the Raspberry Pi 3 development board as the hardware platform for testing. It is one of the boards recommended by the OP-TEE developers [32]. This board includes quad ARM Cortex-A53 processors. We perform our evaluation on the OP-TEE platform with Linux as our rich OS and OP-TEE as our trusted OS. The reason why we choose OP-TEE is because it is completely open-source, has a very well-maintained code base with clear documentation, and includes an exhaustive test suite, which is utilized to evaluate our attack performance.

We cross-compile our CA and TA from X86 Linux host to AArch64 Raspberry Pi 3. We first write and edit the TA and Makefile code in the OP-TEE TA root directory. The CA code is also written and put in the same directory. Next, we cross-compile both CA and TA code for the entire project from the X86 Ubuntu machine. After that, the TA and CA code is compiled and the binary is integrated into rootfs using buildroot. Once the compilation succeeds, it allows us to encrypt and decrypt some security files.

#### 2.2.1. Synchronous Access Attack Model

Synchronous access attack model starts with calling the dedicated CA program in the normal world. The CA program calls RSA cryptographic operation that is embedded in the TA program running in the secure world. Cryptographic operation uses the secret (key). Our attack inserts a coding procedure into the RSA cryptographic operation to encode the leaked secret with a specific slice of computation with a specific PMU signature. When the encryption and the secret key coding finish, the TA procedure sends back the result to the CA. The CA program then learns the secret by applying decoding machine learning procedure to the PMU signature. In this process, the secret information in the secure world is leaked through the measurements of PMU. A machine learning classification decodes the leaked information.

Specifically, the procedure embedded in the TA is a simple matrix multiplication program listed below. With input *N*, the program multiples a 2×N matrix *A* with another N×2 matrix *B* and the result is stored in a 2×2 matrix *C*. Here, the integer *N* is the encoded RSA secret key value fragment ranging from 0 to 1024 which is accessible to the TAs running in the secure world. To exfiltrate the entire secret key, the encoding procedure takes more than 12 rounds depending on the RSA key’s length.

In the normal world, we access PMU perf_events in user space without kernel privilege, which serves as the receiver. The sender is a secure world application. Running in the kernel space enables it to send information that is not available for user space processes. As shown in Figure 2, we have inserted a decoding program into the TA part of the application whose execution time depends on the leaked value and length of private key fragment. That means the user space PMU counter readings will be affected by the information we want to steal.

Given the data or measurements collected by the PMU counters, we build a machine learning model to estimate the secret key. To construct an accurate machine learning model, we have to run the procedure multiple times with various inputs, which is called the training stage. Let *x* denote the measurements read by the PMU counter resulting from *y* secret fragment encoding. Then a ML model is to construct a mapping from *x* to *y*. Accordingly, the encryption and encoding procedure must be called multiple times with varied inputs in the normal world, since the inputs are only known in the normal world and there is no world switching cost. The code used in the training stage is summarized in Figure 3. Once we have enough training data, that is, the (x,y) pairs, we are able to build an accurate ML model that is further used to estimate the secret key in the testing stage. When the procedure is called in the secure world, the readings collected by the PMU counter also record the world switching cost; therefore, we preprocess the test data by taking off the offset of the switching world cost. The ML model is then applied to the PMU vector so as to estimate the secret key.

#### 2.2.2. Semi-Asynchronous Access Attack Model

Synchronous model captures the PMU readings that are generated more or less from the secret encoding computation. There is always a possibility that only one of the TA initiation or termination ends is synchronous with the CA in the normal world. For instance, the TA could be initiated through an API call making the initiation end synchronous. PMU can be started synchronized with this action; however, once the encryption ends, perhaps the encrypted value is sent out to a third party through a network interface. Hence, when the encryption and the corresponding encoding step ends, it cannot trigger a synchronized read of PMU vector in the normal world. This is the semi-synchronous model.

We can run the TA application in the normal world to estimate a distribution of time from when it is started to when it completes. The mean of this time could be used to setup a timer in the normal world which can trigger the PMU vector read, potentially closely aligned with the end of the encoding procedure. Note that the fidelity of the read PMU vector, and the corresponding machine learning decoding depends on the variance in the TA procedure time. The algorithm and details are shown in Figure 4. In this model, additional noise from computation activities other than the encoding procedure is present, potentially making this model less accurate with less leakage bandwidth.

#### 2.2.3. Asynchronous Access Attack Model

Now imagine that although the TA is invoked by the CA through a synchronized call, the non-determinism exists in the computation activity preceding the actual encryption and encoding. Similar non-determinism may exist at the termination end of TA as well. Hence, neither end needed for PMU trigger is predictable. This is the asynchronous model.

We build the asynchronous model in the following two steps. The non-deterministic noise preceding the encryption is captured by inserting a random timing function from the PMU initiation step to the actual call to the TA function in the CA. The random function is the same as the encoding program to multiply a 2×N matrix with a N×2 matrix with a randomly chosen *N*. By setting the upper limit of *N*, this setup can add extra randomness to the PMU measurements. The details of the asynchronous model algorithm are shown in Figure 5.

## 3. Results

To construct the training data, we choose six different events from the perf_events of PMU counter readings. In particular, we use all six PMU counters in Raspberry Pi 3 and they are assigned with perf events as follows:PERF_COUNT_SW_CPU_CLOCK,PERF_COUNT_HW_CACHE_L1I,PERF_COUNT_HW_BRANCH_MISSES,PERF_COUNT_HW_CPU_CYCLES,PERF_COUNT_HW_BUS_CYCLES,PERF_COUNT_HW_CACHE_RESULT_MISS.

These perf events are selected because their values are affected most by a change in parameter *N* derived from the secret key. The encoding program takes some *k*-bit fragment of the key at a time corresponding to a value *N* used in the matrix multiplication encoding procedure. Larger the *k*, larger the computation time of the encoding procedure, proportional to $2^{k}$ dot products of size $2^{k}$. Each encoding step encodes *k* bits to be transmitted. This increased encoding overhead resulting in lower covert channel throughput/bandwidth. In particular, the prespecified size of the training samples *k* should be at least 1. The best throughput in synchronous access attack model is achieved when *k* ranges from 7 to 10. The reason why we choose not to have *k* go over 10 bits is because the higher throughput leads to a higher risk of detection. If it is less than 10 bits, then the TA stack size just needs 16*1024 bytes = 16 KB to complete the attack. As a result, the risk of detection over the network is reduced vastly. As for the semi asynchronous and asynchronous attack models, we only set the covert channel capacity *k* to be 7 bits in order to ensure the achievable throughput based on channel noise. Since the world switching cost between normal and secure worlds has a small variance, by analyzing the experimental results, we can estimate a constant value offset to normalize the test data collected with the world switching cost for the ML classification based on no world switching cost. The world switching cost *c* is estimated in the units of the six PMU counter values. The world switching cost *c* is calculated as follows:(1)$$\begin{array}{c} c:=\frac{\sum_{i=1}^{n}a_{i}}{n}-\frac{\sum_{i=1}^{n}b_{i}}{n}, \end{array}$$
where $a_{i}$ is the PMU counter value captured in the secure world with included world switching cost, $b_{i}$ is the PMU counter value captured in the normal world without a world switching cost, and *n* is the number of captured PMU vectors. Since the switching world cost is non-negative, the value of the cost *c* is always non-negative.

We use linear regression and SVM machine learning models in the Python sklearn library. We also apply PCA for data pre-processing so as to reduce the variance and memory storage. The details for machine learning models can be found in Section 4.1. After comparing training and testing accuracy for both models, it turns out the test results are quite similar with linear regression, or SVM, or whether PCA preprocessing is used or not used. In support of this observation, we include the results for capacity *k* of 7 bits with asynchronous model in Table 1. The reason why this study chooses to show the testing accuracy of three different machine learning models with the asynchronous attack model is that it is the model with a higher noise level. Despite the similar performance, the linear regression method uses much less memory and time overhead.

The overall test results for different attack models with different values for fragment size *k* bits are summarized in Table 2 and Table 3. In Table 2, it shows the overall accuracy is above 98.53% for 7 to 10 bits synchronous access attack model. As for the semi-synchronous and asynchronous access attack models in Table 2 and Table 3, they can achieve a high overall accuracy of 94.52%, and 95%, respectively. When increasing the randomness of the function call for the asynchronous access attack model, the covert channel accuracy is reduced to 83.91%. The bandwidth/throughput of our covert channel can be as high as 11.701 KB/s in synchronous access attack model, 8.140 KB/s in semi asynchronous access attack model, and 8.212 KB/s in asynchronous access attack model. Though Cho et al. also use PMU counting cache refill events to build a covert channel, the proposed covert channel has much lower noise level. As a result, we achieve much lower error rates. In particular, error rates are reduced by 1.47% in synchronous model and 5.48% in both semi-synchronous and asynchronous models. While Cho et al. achieves bandwidth of 95 B/sond (under the cross-core scenario), we can achieve bandwidths as high as 11.701 KB/s. Moreover, our proposed attack can be performed in user space without kernel privilege, since the six chosen perf events are all included in user space perf_event_open system call, whereas “L1/L2 cache refill” perf event used by Cho et al. requires kernel privilege.

## 4. Discussion

### 4.1. Machine Learning Models

One of the common machine learning problems is to find a good function mapping *f* from the input *x* to the output *y* based on the given training sample $S:={\left\{\left(x_{i},y_{i}\right)\right\}}_{i=1}^{n}$. This is called supervised learning since the output or label *y* in the training sample *S* is supervised. In general, we cannot search through all possible functions to find a good function *f*. Instead, we restrict our search within a function class called hypothesis class *H*. Specifically, we try to find the best function $f^{*}\in H$ that fits the training sample. This strategy is called risk minimization and it can be formulated in an optimization problem as follows:(2)$$\begin{array}{c} f^{*}\in {\mathrm{argmin}}_{f\in H}\frac{1}{n}\sum_{i=1}^{n}\ell \left(f\left(x_{i}\right),y_{i}\right) \end{array}$$
where *ℓ* is a loss function that is selected by the designer. The prediction performance would be much different based on the choice of machine learning model and the hypothesis class *H*. In the following subsections, we introduce two widely used machine learning models, that is, linear regression and support vector machine, and convert our problems into machine learning problems.

#### 4.1.1. Linear Regression

As introduced before, the choice of hypothesis class *H* determines the prediction performance of machine learning problems. The simplest hypothesis class is *linear model*, that is, the hypothesis class *H* is a collection of linear functions. Since the output *y* in our case is secret values that is continuous value, the problem can be formulated as a *linear regression* problem. By using scikit-learn’s linear regression Python package, we can easily train our model as follows:(3)$$\begin{array}{c} reg=LinearRegression().fit(X,y). \end{array}$$

Here, *y* as a vector is the collection of all secret values that is the parameter N derived from the secret key, and *X* is the collection of PMU counter values captured in the normal world, whereas X is for the PMU counter value.

#### 4.1.2. Support Vector Machine (SVM)

Another widely used linear model hypothesis is support vector machine (with linear kernel). In contrast to linear regression, SVM not only minimizes the empirical risk but also maximizes the margin. As a result, SVM may obtain better performance on the test stage by sacrificing a small loss on the training stage.

As a comparison, we also use linear support vector regression. It is also known as liblinear to with scikit-learn’s library SVR Python package as follows:(4)$$clf=make\_pipeline(StandardScaler\left(\right),SVR\left(kernel{=}^{\prime }linear^{\prime },C=1.0,gamma{=}^{\prime }auto^{\prime },epsilon=0.2\right))$$
where kernel is the linear type; c is the regularization parameter, which is default 1.0; Gamma is the kernel coefficient and is set to auto; Epsilon is the epsilon-SVR model, which is 0.2. It specifies the epsilon-tube within which no penalty is associated in the training loss function with points predicted within a distance epsilon from the actual value.

#### 4.1.3. Principal Component Analysis (PCA)

To reduce the usage of memorization, we pre-process the collected data by principal component analysis (PCA). PCA as a powerful dimensional reduction method [33,34] changes the basis of the data set based on their variance so that one can only use a few principle components and ignore the rest. As a result, the original data are projected onto a lower-dimensional space and the expense of memorization is reduced significantly and the prediction performance could be also improved.

We first standardize the features into the same scale use scikit-learn StandardScaler() class which is the preprocessing submodule in scikit-learn. Then we use the scaler object’s fit() method with the input X to calculate the mean and standard deviation for each variable in the dataset. Where X is the PMU counter values. Finally, we perform the transformation with the transform() method of the scaler object. The scaled values of X are stored in the variable X_scaled. The procedure is as follows:(5)$$\begin{array}{c} scaler=StandardScaler() \end{array}$$
(6)$$\begin{array}{c} scaler.fit(X) \end{array}$$
(7)$$\begin{array}{c} X\_scaled=scaler.transform(X) \end{array}$$

Then we apply PCA() to fit the scaled values of *X* and apply the transformation. The n_component is set to 0.95, which means preserving 95% of the variance amounts to the principal components.
(8)$$\begin{array}{c} pca=PCA(n\_component=0.95) \end{array}$$
(9)$$\begin{array}{c} pca.fit(X_{scaled}) \end{array}$$
(10)$$\begin{array}{c} X_{pca}=pca.transform\left(X_{scaled}\right) \end{array}$$

As we summarized in Table 1, after comparing the above mentioned machine learning models, they provide minimal difference in prediction accuracy. For the consideration of space and time cost efficiency, we adopt linear regression model for the rest of the data processing.

### 4.2. Limitations & Mitigation Strategies

There are four main limitations and corresponding mitigation strategies that might be considered to restrict the proposed covert channel: PMU counters read, noise-injection, disabling, and isolation. They are summarized below.

PMU counters read is an expensive operation that affects our covert channel’s bandwidth. This study uses syscall perf the Linux profiler, which also calls perf_events to record performance counters. It provides an integrated framework for performance analysis, which can record different cores performance counters from user space. The main limitation of perf is the performance overhead. It impedes us from building a higher bandwidth convert channel. To overcome the perf syscall’s overhead, the approach is to access PMU counters directly with assembly code; however, in cross-core scenario, this approach is not reliable since different cores do not share PMU counters. A corresponding mitigation technique could be to add additional delay/wait in perf counter read functions.

Noise-injection is another factor impacting the performance of the covert channel. The uncertainty in what code is executed within the TA through orthogonal code execution resulting from interrupts increases the variance in the PMU counters values. A high variance in the TA code execution with embedded PMU events results in high variance in the PMU values. This reduces the accuracy of the decoding ML procedure. Additional random orthogonal code that affects the relevant PMU event counters could be scheduled as a mitigation technique.

Disabling the vulnerable PMU counters is an efficient solution to defend against PMU-based attacks. However, it will compromise software developers’ ability to utilize PMU counters to analyze and improve the performance of their code. Moreover, this study has not tested the covert channel techniques on other hardware platforms, which provide different set of events monitored by the PMU counters. In other words, if there are PMU counters monitored events that have not been disabled, our covert channel still be effective.

Isolation-based approaches prevent processes in different trust domains from sharing hardware. They are highly effective at mitigating PMU-based cross-world covert-channel attacks. Such hardware isolation can be achieved by using hardware partitioning techniques that eliminate cross-world transmission through this covert channel. If the TEE such as Intel SGX also provides isolation for performance monitoring events associated with a confidential enclave, this PMU-based covert channel is not feasible; however, even SGX supports uncore PMU counters which measure various events that occur outside the actual cores. Uncore PMU counters monitor events related to power management, IO bandwidth, memory bandwidth, interconnect traffic, and other metrics. By testing vulnerable uncore PMU counters, there is a good chance that our cross-world covert channel can succeed on SGX enabled platforms as well.

## 5. Conclusions

In this paper, we build and evaluate a covert channel between the ARM TrustZone secure world and normal world. A secret within the secure world is exfiltrated to the normal world through an unmonitored PMU. An encoding program encodes a secret *S* as a vector V(S) of some PMU events. The decoding program at the normal world end infers *S* from V(S). In existing PMU event-based covert channels, the PMU events used in the exfiltration vector V(S) are based on cache events such as L1C misses or LLC misses. In order to generate such cache events in the encoding program in the secure world, significant time is needed. The cache level events typically have latency of the order of 100–1000 CPU cycles. We, instead, propose to use much faster microarchitecture level events to encode the secret. This can lead to 100× higher covert channel bandwidth.

A key contributor of noise between secure world client and normal world application covert channel is degree of synchronization between them. If the PMU event read in the normal world can occur right after the PMU based encoding is completed in the secure world, the machine learning inference model can be less noisy and more robust. We propose three different covert channels with different degrees of synchronization to attack TrustZone architecture by stealing the sensitive information in the secure world. This sensitive information is only accessible to trusted applications (TAs) running in the secure world. The leaked information is encoded by the computational work performed which is captured through PMU counters based on microarchitecture level events. The normal world client decodes the leaked information by using a machine learning model based on the PMU vectors. We are able to estimate the leaked information with high accuracy of the order of 95%. We illustrate the performance of the proposed attack models with the accuracy and channel throughput metrics. Based on the experimental results, the leaked information is recovered with high accuracy. The accuracy of the key bits inference is 95% or higher with the covert channel throughput approaching 11 Kbits/s. The traditional cache event based covert channels have shown throughput of 760 bits/s.

We believe that the pipeline centric microarchiecture level event based footprint should lead to 100× bandwidth covert channels compared to the 760 bits/s from the previous work. In this paper, we achieve only a 15× bandwidth improvement. Note that the encoding module based on matrix multiplication still needed two cache level events, PERF_COUNT_HW_CACHE_L1I and PERF_COUNT_HW _CACHE_RESULT_MISS. Future work can explore encoding code fragments where the major footprint is only in the pipeline events to push up the covert channel bandwidth.

## Figures and Tables

**Figure 1 sensors-22-07354-f001:**
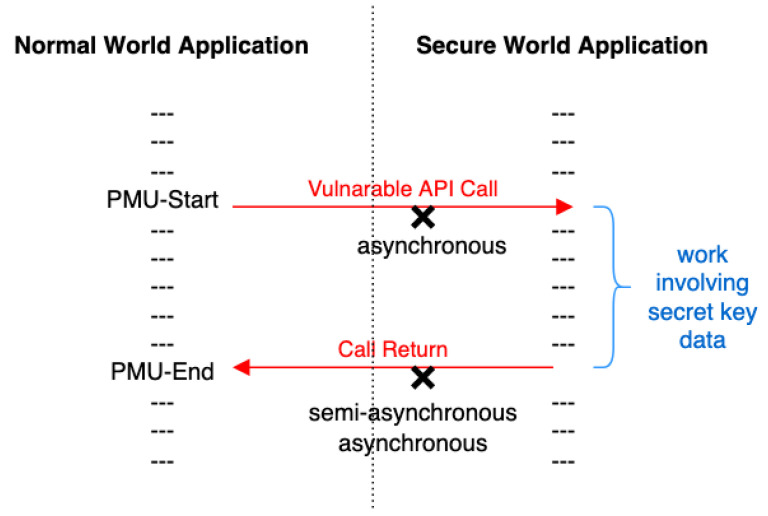
Three attacks: synchronous access attack, semi-asynchronous access attack, and asynchronous access attack.

**Figure 2 sensors-22-07354-f002:**
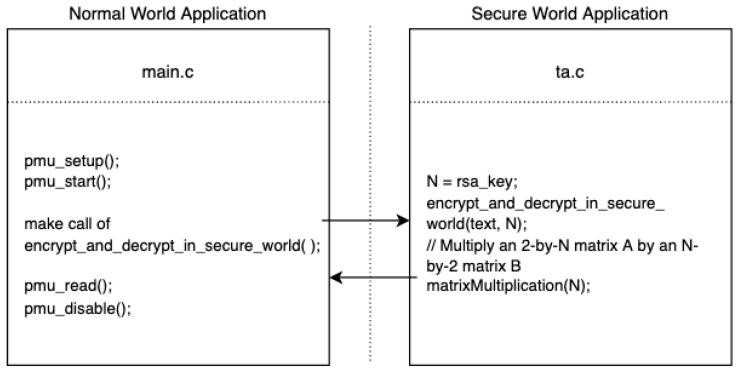
The code used in the synchronous attack model.

**Figure 3 sensors-22-07354-f003:**
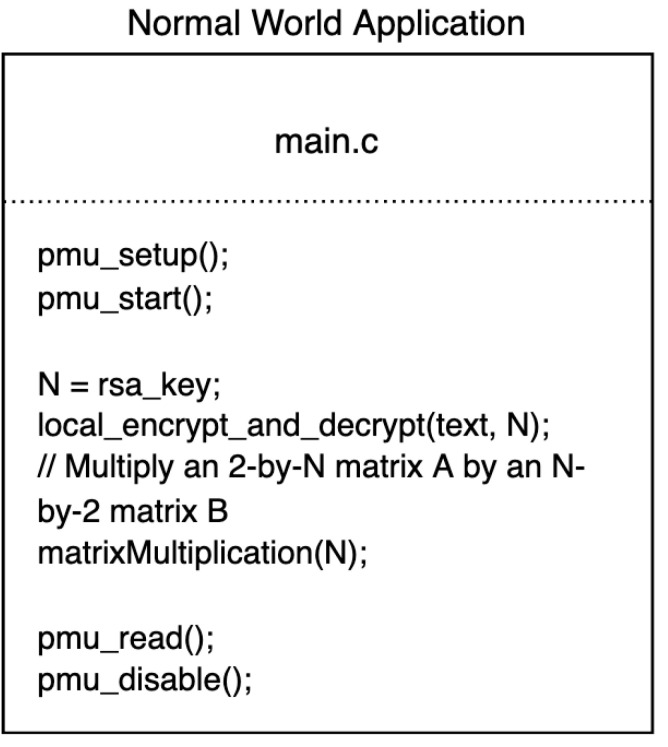
The code applied to the training stage.

**Figure 4 sensors-22-07354-f004:**
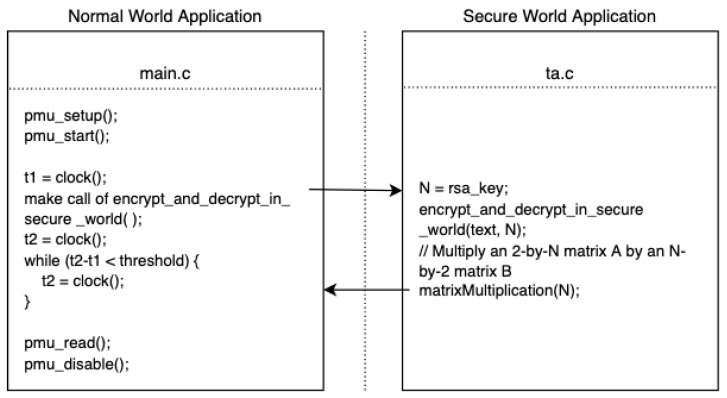
The code used in the semi-synchronous attack model.

**Figure 5 sensors-22-07354-f005:**
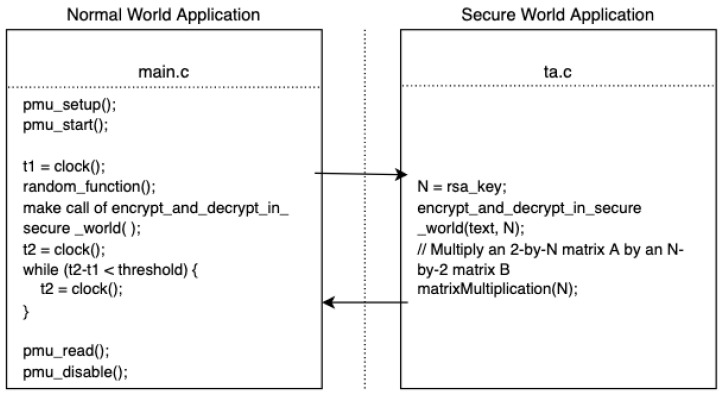
The code used in the asynchronous attack model.

**Table 1 sensors-22-07354-t001:** The accuracy of different machine learning models with asynchronous attack model with k=7 bits: linear regression, SVM, and PCA + SVM, where range is the input parameter of the random noise function called before encryption.

Range	Linear Regression	SVM	PCA + SVM
[0,5]	94.30%	95.14%	95.14%
[0,10]	84.96%	88.07%	88.07%
[0,50]	83.91%	85.53%	85.53%

**Table 2 sensors-22-07354-t002:** The results of synchronous and semi-synchronous attack models, where BW stands for bandwidth (KB/s).

Bits	Test Acc.	BW Max	BW Min	BW Ave.
Synchronous attack model				
7	98.53%	8.199	4.613	6.649
8	99.18%	9.339	4.696	7.238
9	99.04%	9.295	3.960	7.235
10	99.29%	11.701	3.535	6.556
Semi-Synchronous attack model				
7	94.52%	8.140	5.037	6.603

**Table 3 sensors-22-07354-t003:** The results of asynchronous attack, where BW stands for bandwidth (KB/s), and range is the input parameter of the random function called before encryption.

Bits	Range	Test Acc.	BW Max	BW Min	BW Ave.
7	[0, 5]	95.14%	8.204	4.617	6.593
7	[0, 10]	84.96%	8.186	4.592	6.494
7	[0, 50]	83.91%	8.212	4.193	6.619

## Data Availability

Not applicable.

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
