# Peer review of "Cross-World Covert Channel on ARM Trustzone through PMU"

_sensors, 2022, doi:10.3390/s22197354_

Round 1
Reviewer 1 Report
There are only 12 reference papers, if is better if you add more references
Reviewer 2 Report
The abstract needs to be reduced and add contribution
Inappropriate self-citations by authors
Add new references and include theses suggested references:
1. Ansari, A. S., Jawarneh, M., Ritonga, M., Jamwal, P., Mohammadi, M. S., Veluri, R. K., ... & Shah, M. A. (2022). Improved Support Vector Machine and Image Processing Enabled Methodology for Detection and Classification of Grape Leaf Disease. Journal of Food Quality, 2022.
2. Mustafa, M., & Al-Badi, A. (2022). Role of Internet of Things (IoT) Increasing Quality Implementation in Oman Hospitals During COVID-19. ECS Transactions, 107(1), 2229.
3. Mustafa, M., Alshare, M., Bhargava, D., Neware, R., Singh, B., & Ngulube, P. (2022). Perceived security risk based on moderating factors for blockchain technology applications in cloud storage to achieve secure healthcare systems. Computational and mathematical methods in medicine, 2022.
4. Kassanuk, T., Mustafa, M., & Panse, P. (2021). An Internet of Things and Cloud Based Smart Irrigation System. Annals of the Romanian Society for Cell Biology, 20010-20016.
Make your contribution clear
Reviewer 3 Report
Authors are suggested to uses third form here i.e. ( Don't us we). Problem Statement & objective not cleared in this research work. Latest references would not seems in this paper. Improve the novelty of paper.
Reviewer 4 Report
This paper is proposed a covert channel between the ARM TrustZone secure world and normal world through PMU by three different synchronization capabilities between the client and trusted applications in the covert channel. The authors give a comprehensive analysis on the schemes, but this article has several issues that need to be resolved.
1. Overall, there is a great room of improvement. The novelty of the work needs to be clearly elaborated.
2. Research contribution need be highlighted clearly, not able to find any significant?
3. Discussion section should be extended to present appropriate information about the research topic for the reader especially at the abstract and conclusion.
4. Authors need to specify how the authors benchmarking and validate their proposed method for covert channel and trained through PMU. It is not specify in the abstract and not clear explanation in the paper.
5. Section 1.1.2, there is figure to illustrate about the TrustZone Software Architecture.
6. Section 1.2 Related Work, it’s not promising since authors only focus related work on Reference no 3. It should covers other existing works.
7. Line 180-183, the sentence needs to re-write and explain more details. What is the assumption at line 183?
8. The authors should provide a flowchart for section 2.2. The explanation from line 196-210 are confusing.
9. There are too many used of ‘we’ in the entire of the paper, it would be suggested to use passive sentence.
10. Authors need to include details pseudocodes for each model of Synchronous, Semi-Asynchronous and Asynchronous Access Attack Model.
11. Kindly ensure all the references are recent and relevant.
12. Most of the results should be explained in details and comparative graphical illustrations with detailed data analysis are required for quality improvement of the paper.
13. In results section, how the randomness of the function call has been generated for the asynchronous access attack model? The authors should explain and justify from the findings.
14. What is the novelty in this approach that is giving you the performance advantage needs to be very clearly and if possible through analysis need to be brought out in the paper?
15. The writing is not good enough. There are too many grammar mistakes and typos. It needs careful check.
16. Again, the contributions are not highlighted well in this paper.
